# Effect of Sintering Temperature and Solution Treatment on Phase Changes and Mechanical Properties of High-Nitrogen Stainless Steel Prepared by MIM

**DOI:** 10.3390/ma16062135

**Published:** 2023-03-07

**Authors:** Weipeng Zhang, Liejun Li, Chengcheng Huang, Tungwai Ngai, Ling Hu

**Affiliations:** Guangdong Key Laboratory for Processing and Forming of Advanced Metallic Materials, South China University of Technology, Guangzhou 510640, China

**Keywords:** high-nitrogen stainless steel, metal injection molding, nitrogen concentration gradient, phase transformation, mechanical properties

## Abstract

High-nitrogen stainless steel (HNSS) has been widely concerned and studied owing to its excellent mechanical, corrosion resistance, and biocompatibility properties. A series of HNSS was prepared by metal injection molding (MIM) using gas atomized Fe–Cr–Mn–Mo–0.3 N duplex stainless steel powders. Both sintering and solution treatments were carried out in an N_2_ atmosphere. The effects of nitrogen distribution and phase transformation on the mechanical properties of MIM HNSS during sintering and solution were studied. The results show that as the sintering temperature increased, the sample density increased, but the total nitrogen content decreased. Nitrogen and Cr_2_N concentration gradients along the cross-section of as-sintered samples were formed after cooling. The high nitrogen content promotes the decomposition of γ: γ_saturated_ translated to γ and Cr_2_N. Meanwhile, the low Mn content in austenite also decomposes γ: γ translated to α and Cr_2_N. After solution treatment, a single γ phase was obtained for samples sintered at 1200 to 1320 °C. For solution treatment samples sintered at 1320 and 1350 °C, their tensile strength was 988.76 and 1036.12 MPa; yield strength was 615.61 and 636.14 MPa, and elongation was 42.58 and 40.08%, respectively. These values vastly exceeded the published MIM HNSS values.

## 1. Introduction

High-nitrogen stainless steels (HNSSs) are widely used as structural materials, particularly austenitic stainless steels, which have broad application prospects in the medical industry due to their nonmagnetic characteristic [1], high tensile properties, good wear resistance [2], high toughness [3,4], impressive corrosion resistance [5], and biocompatibility [6]. Nitrogen has a strong effect on forming and stabilizing austenite [7,8] and improving mechanical properties [9] and corrosion resistance [10] of the steel. It is a good substitute for expensive nickel and can be used as a low-cost alloying element to develop commercial HNSS. Solid-phase nitriding is an effective way to prepare HNSS. However, for parts with thick sections, it is difficult for N to penetrate deeply enough in a short nitriding time [11]. Therefore, nitriding is suitable for foils, plates, and thin or porous parts. Metal injection molding (MIM) is an excellent technique for the mass production of small and precision parts [12,13]. The large number of pores formed after debinding can increase the contact area between the nitrogen gas and the powder, thus shortening the diffusion distance during nitriding [14,15,16,17,18]. However, nitridated HNSS typically has a nitrogen concentration gradient along the cross-section of the sintered part. Furthermore, various phases may be formed during the cooling process after sintering, including the Cr_2_N, σ, and χ phases [19,20,21,22,23,24]. A solution treatment followed by rapid cooling effectively eliminates element segregation and the inhomogeneity of the microstructure after nitriding [14]. Qu et al. studied the effects of sintering time and N_2_ partial pressure on the density and nitrogen content of MIM high-nitrogen nickel-free austenitic stainless steel [21,25,26]. However, the nitrogen distribution, phase changes, and element distribution during sintering and nitriding have not been systematically studied.

This study investigated the nitrogen concentration and its distribution, microstructure, and mechanical properties of MIM Fe–Cr–Mn–Mo–N stainless steel sintered and solution-treated under a nitrogen atmosphere. The sintering temperature effect and solution treatment on the phase changes and mechanical properties were discussed.

## 2. Materials and Methods

### 2.1. Specimen Preparation

Spherical N_2_ gas atomized duplex stainless steel powder (ferrite and austenite, with an average particle size of 11 μm and chemical composition of 16–18% Cr, 11–12% Mn, 3–4% Mo, 0.8% Si, 0.3 N, 0.03% C in wt.%, and Fe-balanced) provided by Sandvic Osprey, China was used. The feedstock was prepared by mixing 68 vol.% of the as-received powder and 32 vol.% of the binder, which included 65 wt.% microcrystalline wax, 25 wt.% high-density polyethylene, 5 wt.% ethylene-vinyl acetate, and 5 wt.% stearic acids. The feedstock fabrication process included mixing, granulation, injection, debinding, and sintering; the mixing temperature was 140 °C, and the rotational speed was 30 r/min. Injection molding was performed by using an injection machine (Foshan Liquan LQ-98A, Foshan, China) with injection temperature of 140 °C and injection pressure of 100 bar. Cylindrical samples with a diameter of 25 mm and a thickness of 6 mm were obtained after injection. Two-step debinding was used. First, the microcrystalline wax was dissolved in n-heptane at 62 °C for 8 h, and the remaining binder was removed by thermal debinding under flowing N_2_. The heating schedule was as follows: heating from the room temperature to 410 °C and holding for 1 h, holding for 1 h at 500 °C, and holding for 2 h at 720 °C, then furnace-cooled to room temperature. The debinded samples were sintered at 1200, 1250, 1300, 1320, 1350, and 1380 °C for 2 h under flowing N_2_ in a GSL-1700 tube furnace (Hefei Kejing Material and Technology Co., Ltd., Hefei, China). The sintered samples were named N1200, N1250, N1300, N13 20, N1350, and N1380, respectively. The sample thickness after sintering ranged from 5.2 to 5.5 mm. Solution treatment was carried out at 1150 °C for 1.5 h under flowing N_2_ in a GSL-1200 tube furnace, followed by water quenching, and samples were named GN1200, GN1250, GN1300, GN1320, GN1350, and GN1380, respectively. The heating rate for all the heating processes was 10 °C/min. Figure 1 shows a flow chart of sample preparation.

### 2.2. Characterization of Sintered Specimens

Microstructure investigation was performed using an optical microscope (OM) and scanning electron microscope (SEM, ZEISS, Sigma300, Oberkochen, Germany). The polished surface of sintered samples was chemically etched in aqua regia (75 vol.%HCl-25 vol.%HNO3) for 5–15 s, while for solution-treated samples, the polished surfaces were chemically etched by 10 wt.% oxalic acid solution for 20–30 min. SEM equipped with an energy-dispersive X-ray spectroscope (EDS) was used to analyze the chemical composition. X-ray diffraction (XRD, Panalytical X’pert, PANalytical B.V., Almelo, The Netherlands) operated at 40 kV, and 40 mA with Cu-Kα radiation was employed for phase identification. XRD patterns were recorded with a scanning speed of 12°/min. Samples for TEM analysis (3 mm in diameter) were prepared by grinding the samples to 45 μm thickness, followed by thinning with an argon ion beam. Mn content in different states was analyzed by X-ray Photoelectron Spectroscopy (XPS, Thermo Scientific Escalab 250Xi, Waltham, MA, USA) using an Al-Kα anode X-ray source, operating under a vacuum of ~2 × 10^–7^ Pa. A step size of 0.05 eV was employed, and the diameter of the analyzed area was 400 μm × 300 μm. Data for the binding energy of Mn and N elements were obtained from the NIST XPS database [27] and published papers [17,18,28]. Nitrogen content in the bulk alloys was measured by an oxygen and nitrogen gas analyzer (TC600, LECO, Benton Harbor, MI, USA). The average solution nitrogen content in the austenite phase was estimated based on lattice parameter calculation according to semi-Vegard’s law (α = 0.3587 + 0.0010 [at. pct N]) [19,20]. Cross-sectional polished samples were prepared to detect center zone by XRD, SEM, and XPS.

Tensile properties were measured using a universal testing machine (MTS Test Star 810, MTS Systems Corporation, Eden Prairie, MN, USA). Tensile test specimens with a gauge length of 8 mm, a width of 2 mm, and a thickness of 1 mm were machined from bulk samples after removing the surface layer. Tensile tests were performed at a constant rate of 0.0083 mm/s, corresponding to an initial strain rate of 1 × 10^−3^ s^−1^. Vickers hardness was measured using a force of 0.98 N (HV_0.1_) with a dwell time of 15 s.

## 3. Results

### 3.1. Phase Constituent and Microstructure of the Sintered Samples

Figure 2 displays the XRD patterns of the sintered samples. Figure 2a shows the results on the sample surface. As the sintering temperature increases, the phase constituents change from austenite (γ) and Cr_2_N (sample N1200) to γ, ferrite (α), and Cr_2_N (samples N1250 to N1380), indicating that the α phase begins to appear in the surface layer after the sintering temperature reaches 1250 °C and higher. The diffraction results obtained at the interior of the samples are presented in Figure 2b. The phase constituent at the interior change from γ and Cr_2_N (sample N1200) to single γ (samples N1250 to N1320) and finally change to γ plus ferrite (δ) (samples N1350 to N1380). When comparing Figure 2a,b, the surface and the interior of the sintered samples have different microstructure evolution patterns. The appearance of high-temperature ferrite (δ) was found at 1350 °C in the sample interior. Furthermore, even at the same sintering temperature, the surface and interior phase constituents are different except for sample N1200, which has the same phase constituent of γ and Cr_2_N.

Figure 3 shows the OM and SEM (insert in (a) and (c)) results taken along the cross-section of all the sintered samples. According to the figure, the grain grows gradually as the sintering temperature rises, and the austenite grains have twin structures. For the sample sintered at 1200 °C, there is no obvious difference in the phase constituent along the cross-section of the sample, which contains γ and discontinuous lamellar and intragranular Cr_2_N. For the samples sintered at 1250 to 1380 °C, a surface layer forms along the cross-section of the sample, and their thickness decreases gradually with increasing sintering temperature. The thickness of a surface layer on samples N1320, N1350, and N1380 is about 167, 109, and 88 μm, respectively (Figure 3d–f). Figure 4 shows the SEM microstructure and EDS results obtained from sample N1200. Strip-shaped and irregular lump-shaped Cr_2_N can be found, and the EDS results show that their atomic ratio of Cr to N is about 2:1. Figure 5a shows a bright-field (BF) TEM image of sample N1320. Grain boundaries are discernable. Both lamellar (area marked B) and round precipitates are observed in the microstructure. Figure 5b,c show the EDS spectra from the marked areas in Figure 5a. The EDS results confirm that the lamellar precipitate is rich in Cr and N, while the round precipitate is rich in Mn and O. Figure 6a is the EDS line scanning result along the sample cross-section N1200. The N and Cr concentrations decrease slowly, but the Fe and Mn concentrations do not change much. Figure 6b is the EDS line scanning result along the cross-section of sample N1350. From Figure 6b, Fe depletion and N and Cr enrichment can be found in the surface layer, while the Mn concentration remains constant. Figure 7 shows the N concentration distribution along the cross-section of samples N1200 and N1350. The N content at both ends of the samples is higher than that in the interior. Because the lower surface of the sample was in contact with the crucible, the upper free surface has a higher N content than the lower surface. The average nitrogen content measured by the oxygen and nitrogen gas analyzer in sample N1200 is 1.27 wt.%, and sample N1350 is 0.72 wt.%.

### 3.2. N Concentration Gradient and Microstructure Evolution of Sintered Samples

However, for samples N1250 to N1380, the surface layer contains three phases (Cr_2_N, γ, and α). The appearance of α was caused by the decrease in Mn(met.) and the depletion of N in austenite. Figure 8 depicts the XPS peak convolution of Mn on the surface layer and in the interior of sample N1320. Although the Mn content along the cross-section is consistent (Figure 6b), the percentage of Mn(met.) on the sample surface is lower than that in the sample interior. In comparison, the percentage of Mn-oxide on the sample surface is higher than that in the sample interior because Mn is easily effused to the sample surface and oxidized at a high sintering temperature. The nitrogen content in austenite was calculated based on the lattice parameters, and the result is shown in Table 1. For all samples, the nitrogen content in austenite on the sample surface is lower than in the sample interior. For samples N1250 to N1380, the nitrogen content in austenite on the sample surface gradually decreases from 0.72 to 0.43. However, in the interior of samples N1250 to N1380, the nitrogen content in austenite is above 0.8 wt.%. Two kinds of decomposition of austenite appeared on the sample surface: (1) γ_saturated_ translated to γ and Cr_2_N because of a high N content and (2) γ translated to α and Cr_2_N because of a low Mn(met.) content. However, the austenite does not decompose thoroughly in the sample interior and is retained at room temperature during cooling because of a lower N content and higher Mn(met.) content. Therefore, a microstructure of γ and Cr_2_N for samples N1250 to N1320 was formed in the sample interior. However, γ, δ, and precipitated phases (Figure 3f) in the interior of samples N1350 and N1380 are observed because at that sintering temperature, the N content was insufficient, and the δ [29,30] phase appeared at high temperatures in the interior of the samples and is retained at room temperature after cooling.

Meanwhile, some secondary phases precipitated from the δ phase and appeared at the grain boundary (Figure 3e,f) during cooling. The EDS result shows that they are rich in Fe, Cr, and Mo. The diffraction peaks of those precipitated phases cannot be found in Figure 2, and the Cr_2_N in the interior of samples N1250 to N1380 cannot be identified in Figure 2b due to its low content, which is lower than the detection limit of XRD instrument.

### 3.3. Mechanical Properties after Sintering

The relative density of the as-sintered samples is shown in Table 2. As the sintering temperature increased, so did the relative density, with a high relative density of 98.18% obtained when the sintering temperature reached 1380 °C.

Figure 6 shows the hardness distribution along the cross-section of the samples sintered at 1200 and 1350 °C. The hardness of the sample is mainly affected by the relative density, nitrogen content, and secondary phase [31,32,33]. Hardness data show that the hardness on both surfaces of samples N1200 and N1350 is greater than that on the interior. Meanwhile, the hardness on the upper surface of samples with higher N and Cr_2_N contents is higher than on the lower surface. The maximum difference in hardness between the surface and the interior is 208.33 HV for sample N1350 and 231.66 HV for sample N1380, which is the highest because of the longest nitriding-cooling time, resulting in the maximum content difference of the Cr_2_N precipitation among all samples. The hardness reported in Table 2 is the average of the data points taken at the interior of the samples (about 0.5 mm below the upper and lower surfaces). The relative density in Table 2 was measured by the Archimedes method.

Table 2 shows the tensile properties, and Figure 9a shows the tensile stress–strain curves for the as-sintered samples. The overall tensile properties increased (samples N1200 to N1320) and then decreased (samples N1320 to 1380) with an increasing sintering temperature. The highest tensile strength and yield strength were obtained in sample N1200 due to its fine grains (about 12 μm) and highest nitrogen content, causing significant Cr_2_N precipitation that can strengthen the precipitation strongly [34]. However, these Cr_2_N precipitations cause crack initiation resulting in a low elongation. Meanwhile, sample N1200 has the lowest relative density (about 12% porosity), which reduces plasticity. For samples N1250, N1300, and N1320, the yield strength decreases gradually because the N and Cr_2_N contents decrease. Samples N1320 and N1350 have the best overall mechanical properties among all samples, with a tensile strength of 842.47 and 916.34 MPa, a yield strength of 613.78 and 690.65 MPa, and an elongation of 29.75 and 21.35%, respectively.

Furthermore, when sample N1350 was compared to N1320, the tensile strength and yield strength were higher, but the elongation was lower due to the ferrite (δ) phase, which increases strength but decreases plasticity. Sample N1380 with the δ phase of a large grain size and hard and brittle phases appearing at the grain boundaries (Figure 3f) leads to high yield strength but low elongation. The fracture surfaces of the as-sintered sample subjected to the tensile tests are presented in Figure 10. When sample N1320 was compared to sample N1200, the dimple number on the fracture surface increased, indicating that the plasticity increased. However, as the sintering temperature increased to 1380 °C, the fracture mechanism transformed from a ductile fracture to a brittle cleavage fracture. The excellent fracture toughness of sample N1320 is attributed to the austenite matrix and the absence of the Cr_2_N (N1200), δ, and precipitated phases (N1380). Furthermore, round particles were discovered in the fracture dimples and the EDS energy spectrum. The results reveal that it is rich in Mn and O.

### 3.4. Microstructure after Solution Treatment

Figure 11 shows the XRD results in the interior of samples GN1200 to GN1380. As shown in the figure, the phase constitution is a single γ after solution treatment for samples GN1200 to GN1320, but the phase constitution is γ and δ for samples GN1350 and GN1380. Figure 12 represents the OM results on the cross-section of samples GN1200 to GN1380. Compared to the corresponding as-sintered samples in Figure 3, the layered structure along the cross-section of the as-sintered samples (samples N1200 to N1350) disappears, and the microstructure becomes homogeneous. A single γ with a twin structure can be observed in Figure 12a–e. Although ferrite can be detected by XRD, it cannot be observed in Figure 12e, and this is probably due to the ferrite’s small size and being etched away before observation. In Figure 12f, the δ remains preferentially at the austenitic grain boundary. By comparing the sample grain size before and after the solution treatment (Figure 3 and Figure 12), the grain size remains unchanged. It is worth noting that, in contrast to the matrix, which comprises γ and δ, a single γ phase surface layer with a thickness of about 107 μm can be seen in sample GN1380, as shown in Figure 12f. The high nitrogen content in the surface layer of sample GN1380 encourages the γ formation after solution annealing. However, the relatively low nitrogen content in the sample interior cannot eliminate all the δ after solution treatment. Figure 13 shows the EDS line scanning result along the cross-sections of samples GN1200 and GN1250. The N, Cr, Mn, and Fe concentrations do not change much, and it shows that the solution treatment at 1150 °C for 1.5 h can obtain samples with a uniform microstructure without segregation.

### 3.5. Mechanical Properties of Solution-Treated Samples

Table 2 shows the tensile properties, and Figure 9b shows the tensile stress–strain curves of samples GN1200 to GN1380. The fracture strain of the samples increases after solution treatment. The fracture surfaces of the solution-treated samples subjected to the tensile tests are presented in Figure 10. After solution treatment, the fracture surface has some dimples for samples GN1200 and GN1380, indicating that their fracture toughness was improved. The tensile strength of sample GN1200 is 155.7 MPa, which is lower than sample N1200 because of the elimination of nitrides, and the yield strength of sample GN1200 is 415.8 MPa, which is lower than N1200. The yield strength of samples GN1250, GN1300, and GN1350 is around 620 MPa and close to that of samples N1250, N1300, and N1350, respectively, due to the overall influence of the N content in austenite and the Cr_2_N content. Sample GN1380 has the highest yield strength of 810.1 MPa because it has the highest relative density and a dual-phase microstructure (γ plus δ). δ is a cubic body-centered phase that can inhibit dislocation movement and thus increase yield strength. A tensile strength of approximately 1 GPa was achieved for all samples. For samples GN1200 to GN1320, the elongation increases gradually with an increasing sintering temperature but reverses for samples GN1350 and GN1380. Samples GN1320 and GN1350 have the best mechanical properties among all samples, with a tensile strength of 988.76 and 1036.12 MPa, a yield strength of 615.61 and 636.14 MPa, and an elongation of 42.58 and 40.08%, respectively, which are higher than those previously reported for PM HNSS [15,16,17,18,19,20] and cast high-strength HNSS [34] and are close to those HNSS prepared by hot isostatic pressure [35] or SPS sintering [36].

## 4. Discussion

The results show a nitrogen concentration gradient along the cross-section of sintered samples. Previous work [16] has reported that for MIM HNSS, the N content in the sintered sample will decrease with increasing nitriding at a sintering temperature (1200–1320 °C). As displayed in Figure 6, the results from this work also show the same conclusion that the increase in sintering temperature leads to a nitrogen content decrease. Table 2 shows the relative density of the as-sintered samples. The relative density of the sintered samples increases with temperature, with the sample sintered at 1200 °C having a low relative density of 88% (high porosity) while for samples sintered at a higher temperature, their relative density is much higher, ranging from 92% to 98%. The lower the porosity (or defects), the slower the N diffusion in the samples. When the sample has a large thickness and low porosity, nitrogen cannot penetrate deep into the sample. Therefore, when the chemical composition and sample thickness are nearly the same, the N content and its distribution of the MIM-prepared samples are affected by nitriding temperature and sample porosity. Certainly, a nitrogen concentration gradient will form when sintering at a high temperature (1200–1380 °C). However, this concentration gradient is relatively small because of the faster diffusion rate of N than that of nitriding at a lower temperature. However, because cooling was carried out under an N_2_ atmosphere, the nitriding of the sintered sample did not stop. Therefore, the slow diffusion rate of N at a low temperature and the precipitation of nitrides during cooling will increase the nitrogen concentration gradient along the sample cross-section. So, the N concentration gradient was formed after cooling. Meanwhile, because sample porosity decreased as the sintering temperature increased, the gradient in sample N1320 was larger than in sample N1200 (Figure 6). The surface layer thickness that formed after being sintered at a higher sintering temperature was thinner than in the samples sintered at a lower temperature (Figure 3b–f).

The results of Figure 2 and Figure 3 indicate that the phase constituent varies along the cross-section. This is caused by the difference in (a) N concentration along the cross-section and (b) the chemical state of Mn. Mn is well known to be a γ stabilizer and can promote N absorption in steel. Meanwhile, N in the solid solution significantly affects the γ phase zone. However, previous studies [19,20,21,22,23,24] have shown that high-N Cr–Mn austenitic stainless steel will precipitate nitrides when annealed at 700–950 °C with a certain incubation period (such as the TTT curve).

Moreover, increasing the N content will shift the nose of the TTT curve to the left and favor nitride formation [8,27]. Furthermore, the nitrides precipitate preferentially at the grain boundary, and as the N content increases, intragranular Cr_2_N precipitate will occur [8,26]. For sample N1200, the nitrogen content (from 1.1 to 1.7 wt% along the cross-section, as displayed in Figure 7a) is the highest among all samples, which leads to the highest content of intragranular Cr_2_N precipitation. This is because a high N content will promote the decomposition of the γ_saturated_ phase.

The micropores in all the sintered samples were left after degreasing and became smaller during sintering densification. Although the presence of micropores weakens the binding of particles, it provides a passage for nitrogen into samples. At a high sintering temperature, the surface and interior of samples differ little in their nitrogen content due to the fast diffusion rate. Furthermore, nitrogen absorption on the surface of the sample occurs during the cooling process in the N_2_ atmosphere, and an N-rich layer forms on the surface, thus inducing the phase change of the sample surface during the cooling process.

## 5. Conclusions

High-N stainless steels were fabricated by MIM. The sintering temperature effect and solution treatment on the microstructure and mechanical properties were studied. The results can be summarized as follows:With increasing sintering temperature from 1200 °C to 1380 °C, the sample density increases, but the nitrogen content gradually decreases.Nitrogen content and distribution along the cross-section of as-sintered samples are not homogeneous. After sintering under the N_2_ atmosphere, the subsequent furnace cooling causes continuous nitriding and Cr_2_N precipitation, resulting in a sample surface layer rich in N and Cr elements.High N content promotes austenite decomposition: γ_saturated_ translated to γ and Cr_2_N. Meanwhile, Mn is easily effused on the surface of the sample surface and is oxidized, resulting in the decomposition of γ: γ translated to α and Cr_2_N.Samples with a homogeneous γ microstructure can be obtained after solution treatment at 1150 °C for 1.5 h under an N_2_ atmosphere for samples sintered at 1320 °C or lower. For solution-treated austenitic samples sintered at 1320 °C, its tensile strength is 988.76 MPa; yield strength is 615.61 MPa, and elongation is 42.58%.Solution-treated samples with a dual-phase structure (γ and δ) that sintered at 1350 °C have a tensile strength of 1036.12 MPa, a yield strength of 636.14 MPa, and an elongation of 40.08%.

## Figures and Tables

**Figure 1 materials-16-02135-f001:**
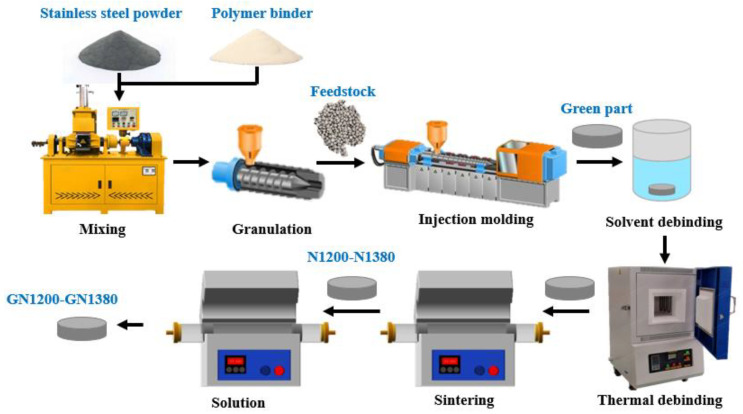
Flow chart of metal injection molding.

**Figure 2 materials-16-02135-f002:**
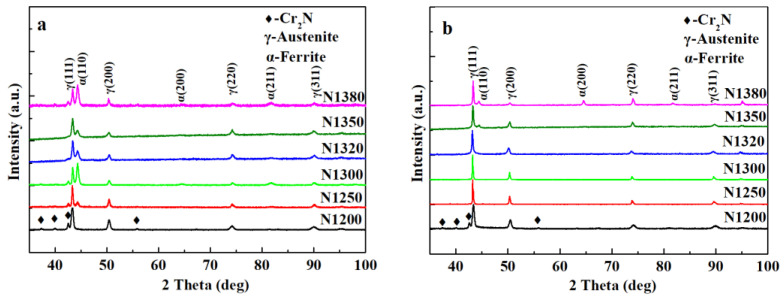
XRD patterns of the samples sintered at 1200, 1250, 1300, 1320, 1350, and 1380 °C (**a**) surface and (**b**) center.

**Figure 3 materials-16-02135-f003:**
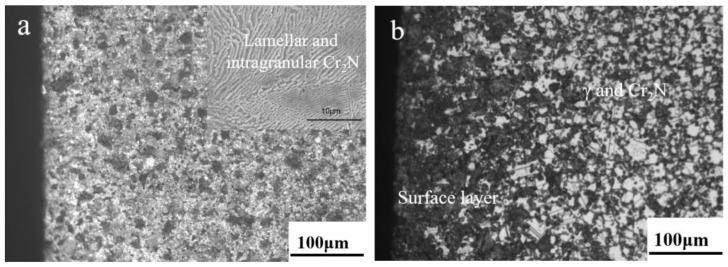
OM and SEM (insert) results along the cross-section of samples, (**a**) N1200, (**b**) N1250, (**c**) N1300, (**d**) N1320, (**e**) N1350, and (**f**) N1380.

**Figure 4 materials-16-02135-f004:**
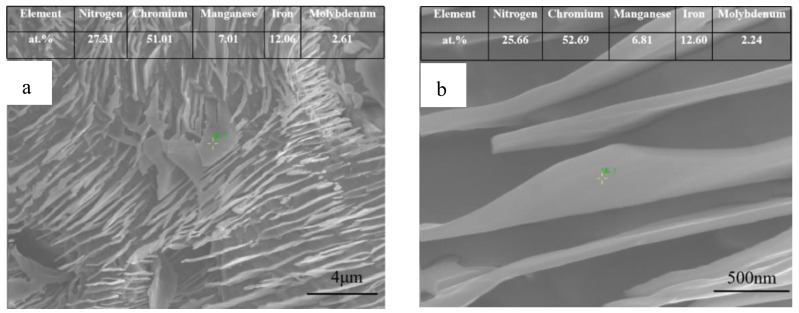
SEM microstructure and EDS results obtained from Sample N1200. (**a**) 5000×; (**b**) 40,000×.

**Figure 5 materials-16-02135-f005:**
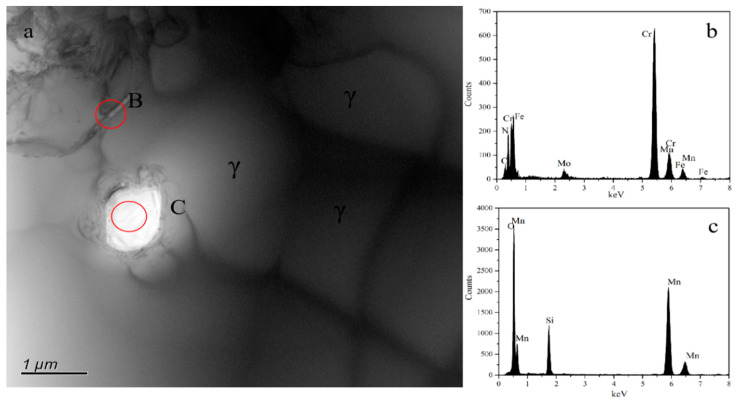
(**a**) BF TEM image of GN1320 sample with 0.72 wt.% N and (**b**,**c**) EDS spectra from B and C phases, respectively.

**Figure 6 materials-16-02135-f006:**
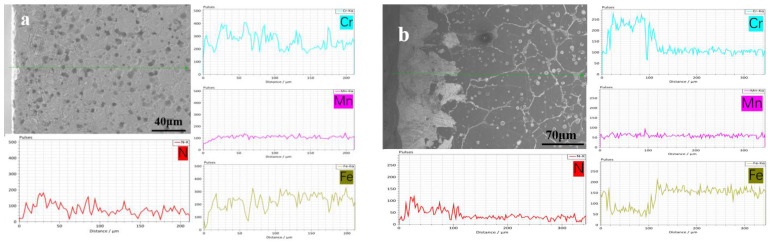
EDS line scanning results along the cross-section of samples (**a**) N1200 and (**b**) N1350.

**Figure 7 materials-16-02135-f007:**
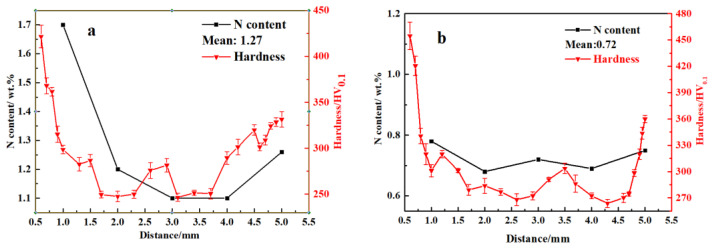
N content and hardness (HV_0.1_) along the cross-section of samples (**a**) N1200 and (**b**) N1350; *X*-axis represents the distance away from the upper surface of samples.

**Figure 8 materials-16-02135-f008:**
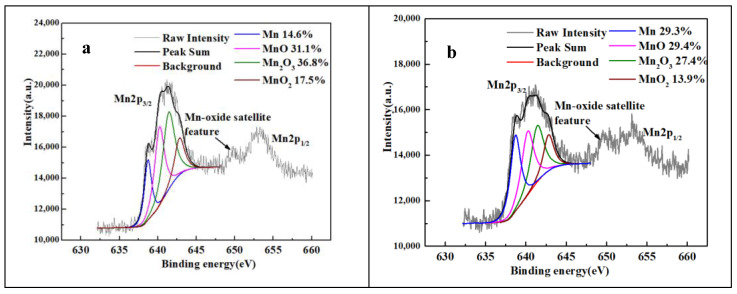
XPS peak convolution of Mn, MnO, Mn_2_O_3_, and MnO_2_ in N1320 sample: (**a**) surface and (**b**) interior.

**Figure 9 materials-16-02135-f009:**
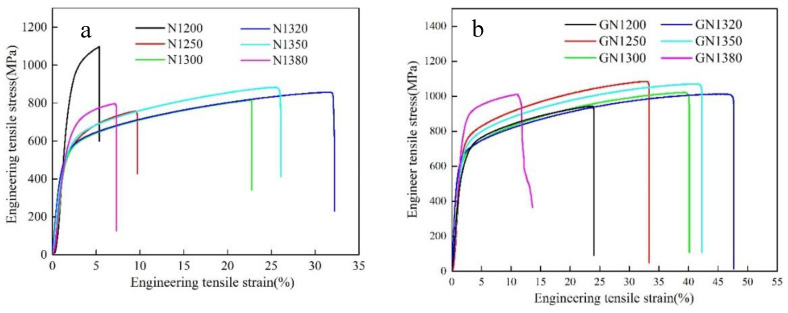
Engineering tensile stress–strain curves of (**a**) sintered and (**b**) solution-treated samples.

**Figure 10 materials-16-02135-f010:**
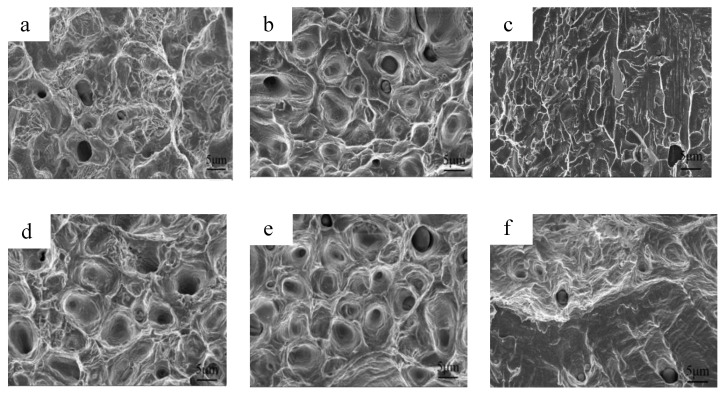
Fracture morphology of the sintered and solution-treated samples (**a**) N1200, (**b**) N1320, (**c**) N1380, (**d**) GN1200, (**e**) GN1320, and (**f**) GN1380.

**Figure 11 materials-16-02135-f011:**
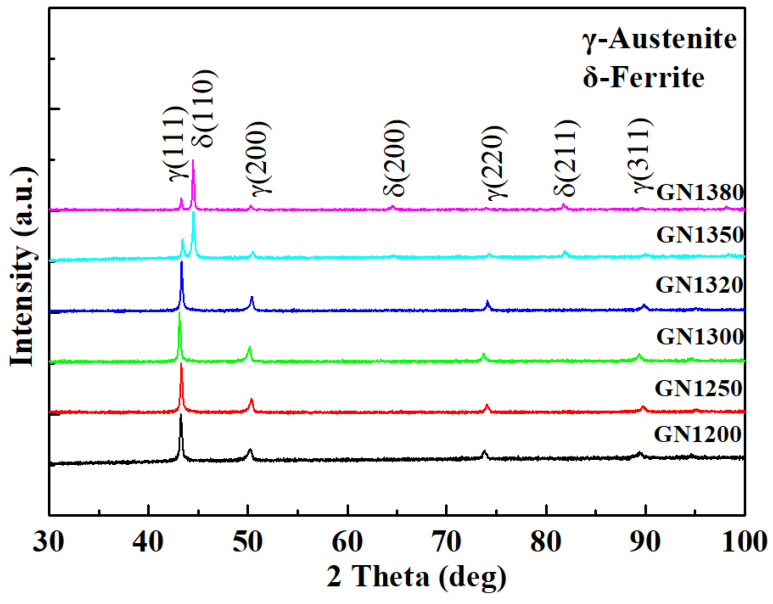
XRD patterns of samples GN1200 to GN1380.

**Figure 12 materials-16-02135-f012:**
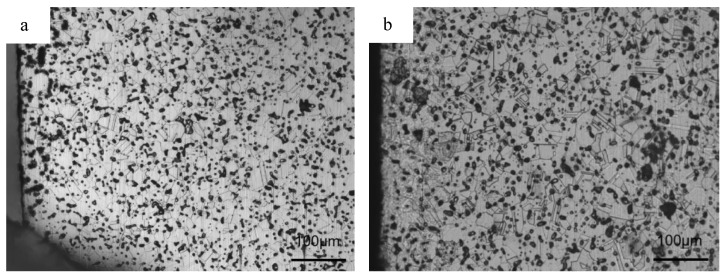
OM results along cross-section of samples (**a**) GN1200, (**b**) GN1250, (**c**) GN1300, (**d**) GN1320, (**e**) GN1350, and (**f**) GN1380.

**Figure 13 materials-16-02135-f013:**
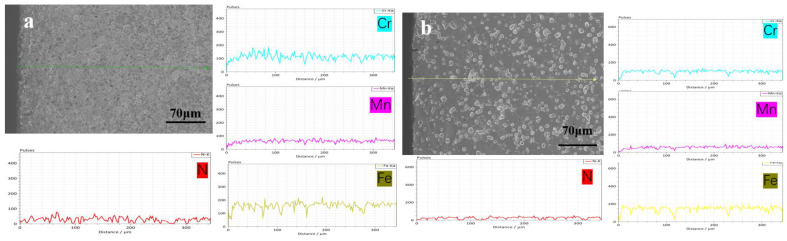
EDS line scanning results along the cross-section of samples (**a**) GN1200 and (**b**) GN1350.

**Table 1 materials-16-02135-t001:** N content in austenite of sintered sample obtained based on lattice parameter calculations.

Sample	N1200	N1250	N1300	N1320	N1350	N1380
Surface/wt.%	0.68	0.72	0.67	0.62	0.57	0.43
Interior/wt.%	0.83	1.01	1.05	1.11	0.92	0.83

**Table 2 materials-16-02135-t002:** Relative density and mechanical properties of as-sintered and solution-treated samples.

Sample Code	Relative Density	Hardness	σ_b_	σ_0.2_	Elongation
	(%)	(HV)	(MPa)	(MPa)	(%)
N1200	88.71 ± 0.8	271.76 ± 30	1113.8 ± 30	1087.3 ± 50	3.4 ± 1
N1250	92.82 ± 0.5	296.23 ± 35	804.7 ± 50	689.5 ± 30	9.5 ± 2.3
N1300	94.41 ± 0.3	272.38 ± 28	835.6 ± 42	613.8 ± 25	18.4 ± 1.6
N1320	95.87 ± 0.5	239.54 ± 32	842.5 ± 28	588.1 ± 26	25.8 ± 3.2
N1350	97.61 ± 0.6	291.39 ± 38	916.3 ± 38	690.7 ± 23	19.4 ± 1.8
N1380	98.18 ± 0.9	309.26 ± 40	804.7 ± 46	709.2 ± 18	5.1 ± 1
GN1200	---	---	958.1 ± 50	671.5 ± 30	17.5 ± 1
GN1250	---	---	1006.9 ± 80	635.4 ± 40	25.5 ± 1
GN1300	---	---	1033.7 ± 30	615.5 ± 10	37.1 ± 2
GN1320	---	---	988.8 ± 30	615.6 ± 40	42.6 ± 3.5
GN1350	---	---	1036.1 ± 30	636.2 ± 60	40.1 ± 3.2
GN1380	---	---	1021.2 ± 50	810.1 ± 10	12.1 ± 1

## Data Availability

All data and charts are presented in the manuscript.

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
