# Peer review of "Effect of Sintering Temperature and Solution Treatment on Phase Changes and Mechanical Properties of High-Nitrogen Stainless Steel Prepared by MIM"

_materials, 2023, doi:10.3390/ma16062135_

Round 1

Reviewer 1 Report

line 190: Figure 8 depicts XPS... correct is: Figure 7 depicts XPS

line 295: Fig. 11: two pictures are marked with the symbol d

line 300: Figure 8a shows ... correct is Figure 8b shows

Author Response

We are grateful for the reviewers’ assessment of our work and deeply apprecciate their thoughtful comments and constructive suggestions. Our point by point reply contains reviewer’s comments in blue, answers in black, and changes in the manuscript in red.

line 190: Figure 8 depicts XPS... correct is: Figure 7 depicts XPS

line 295: Fig. 11: two pictures are marked with the symbol d

line 300: Figure 8a shows ... correct is Figure 8b shows

Reply: It has now been modified in the revised manuscript.

Reviewer 2 Report

In this manuscript, authors show ‘Effect of Sintering Temperature and Solution Treatment on 2 Phase Changes and Mechanical Properties of High Nitrogen 3 Stainless Steel Prepared by MIM’. They present nice spectrum of microstructure, & mechanical properties. The results are very interesting and the paper is generally well prepared. However, in my opinion there are some points that authors should change to improve the paper quality. During its careful review a series of problems and suggestions (listed below) arose.

My recommendation is: Minor revision.

Comments

1.      Please mention the schematic representation of the process.

2.      Scales of figures (6 & 7) should be in bold letters. They are not clearly visible.

3.      Mention the role of micro-porosities of the sintered samples.

4.      English expression needs to be improved. The manuscript contains some grammatical errors and unreasonable sentence structure?

Author Response

We are grateful for the reviewers’ assessment of our work and deeply apprecciate their thoughtful comments and constructive suggestions, which we have fully addresed in this revisison and believe that this has strengthened the manuscript significantly. Our point by point reply contains reviewer’s comments in blue, answers in black.

Comments

  1. Please mention the schematic representation of the process.

Reply: We have added the schematic representation of the process in Figure 1.

  1. Scales of figures (6 & 7) should be in bold letters. They are not clearly visible.

Reply: Figure 6 and 7 have been modified in the revised manuscript.

  1. Mention the role of micro-porosities of the sintered samples.

Reply: we have added the role of micro-porosities of the sintered samples in the discussion.

  1. English expression needs to be improved. The manuscript contains some grammatical errors and unreasonable sentence structure?

Reply: The paper have been fully proofread.

Reviewer 3 Report

Dear Authority,

The manuscript entitled ‘Effect of Sintering Temperature and Solution Treatment on 2 Phase Changes and Mechanical Properties of High Nitrogen Stainless Steel Prepared by MIM’ presents sintering process of duplex stainless steel powders under nitrogen atmosphere. The authors aim to understand the effect of sintering temperature on the microstructural and mechanical properties of sample produced by MIM application. According to the sintering temperature are mainly dominant on the formation of intermetallic phase (Cr2N) and polymorphic phases of iron. The solution treatment, after sintering process, make contribution on the mechanical properties of samples.

Overall, the paper submits some experimental data about sintering behavior of duplex stainless steel produced MIM process. However, there are important points which does not make sense in scientific perspective. Please revisit the manuscript according to following comments;

1- If you aim to sinter the stainless steel samples by MIM application, it is well known that the hydrogen atmosphere needs to be used for sintering process with tungsten or molybdenum furnaces. On the other hand, some people use graphite furnaces with nitrogen atmosphere up to 600 oC and rest of the sintering process is done under Argon atmosphere. But, the quality of sample is still in doubt. If you use Nitrogen gas atmosphere, the intermetallic phase formation (Cr2N) as needle like or elongated grains is inevitable and this issue deteriorate the endproduct quality. So, what is the aim or motivation using nitrogen atmosphere for this study. Otherwise, the quality of the endproduct are questionable.

2- The sintering temperature range for MIM production is very narrow. If you choose 1200 oC for sintering, the samples cannot be sintered even if the sample is kept in furnace in infinite time at this temperature. Because, after dewaxing/catalytic debinding and secondary debinding process, the porous in samples are too big and regular sintering temperature number are not enough for proper sintering. So, there would be always porous structure in sample and nitrogen diffusion deep into sample would be certain. On the other hand, higher sintering temperature could also cause melting of sample, distortion on the shape of sample and grain coarsening which effect the mechanical properties of sample. The suitable sintering temperature range for this type sample is 1320-1340 oC. The preference for sintering temperature range needs to be testified.

3- According to xrd examination, ferrite or austenite phase composition is varied in surface and interior sample at certain temperature range. This issue is mainly related the differences of cooling rate at surface and interior of sample. Please revisit that part again.

3- There are minor mistakes also present in the manuscript;

a) Please provide MIM machine brand information.

b) correct the binder composition of ‘5 vol.% ethylene-vinyl acetate’ as ‘5 wt % ethylene-vinyl acetate’.

I am disappointed to state that this manuscript has serious flaws and blind points so that it is not suitable for publication in Materials.

Best wishes,

Reviewer 4 Report

This paper investigated the mechanical properties of synthesized series of high-nitrogen stainless steel (HNSS) prepared through metal injection molding 10 using gas atomized Fe-Cr-Mn-Mo-0.3N duplex stainless steel powders. Several results are presented, and interpretations are less consistent. I, therefore, recommend significant revisions before publication in materials.

Abstract:

-          Authors may state the aim of the research.

-          Some grammatical errors like in line 12 “….was carried out in an N2 atmosphere” an before N2 should be deleted

-          Line 12, ….The results showed that as….” Comma should be added after that

-          “These values vastly exceed the published…” exceeded not exceed

Introduction

-          Line 27 “…..which have broad application prospects in the medical 26 industry due to their nonmagnetic characteristic” what connected magnetic property with medical application?

Materials and methods

-          Divide this section into subsections to tally that of results and discussion section. You may refer to Metals 20199(7), 755; https://doi.org/10.3390/met9070755

Results and Discussions

-          Fig. 1 not levelled (a & b) as stated in the text

-          JCPDS number of each phase in Fig. 1 may be stated

-          Peaks plane angles of each phase maybe stated

-          Scales in Fig. 2 b-f are not clear

-          Fug. 6 error bars of the hardness values maybe included

-          Figs. are not properly cited in the text. E.g. Figs. 2e, f are mentioned in line 212 while the Figs are presented in other section.

-          Instead of section 3.2, better separate Results and Discussions sections

-          Fig. 10  JCPDS number of each phase may be stated. Peaks plane angles maybe shown

Conclusion

Use bullet instead of numbering.

General:

-          The paper need through proofreading. A lot of grammatical errors

-          Sections and Figures need to be rearranged 

Reviewer 5 Report

Review Paper

The manuscript reports MIM of high nitrogen austenitic alloys, which if they clarify their new contribution is an interesting topic for research.

They show interesting results, have made an extensive investigation in the effect of sintering temperature and solution heat treatment in nitrogen profile.

The paper is well written; however, the introduction and discussion of results can be improved to bring new reference and better discussion.   

After review I believe the manuscript can be published.

I have some comments and question about the manuscript:

·         Which is exactly the novelty of the study. Studies about MIM of   Fe-Cr-Mn-Mo-0.3N have already been published for instance https://doi.org/10.4028/www.scientific.net/AMR.129-131.886)  and DOI: 10.1007/s12613-010-0335-3

The authors should explain in the introduction which is the novelty of their work.

·         The introduction is too short and mot update. In the introduction should be add more recent references.

·         “Feed”, usually it used the term feedstock for MIM feedstocks.

Results and Discussion section

·         What is the meaning subsubsection?

·         How XRD of samples interior was made? Is it a XRD of samples cross section? This should be explained in the methodology. Was the surface polished?

·         Why are the phases different in the surface and in interior of the samples? The authors do not discuss these results.

·         Is there possible to have a phase transformation during sample cutting and polishing due to plastic deformation? 

·         The authors should add reference for their identification of phases (XRD PDFs or articles).

Figure 1:

·         the peaks are overlapping, which makes difficult to analyze their intensity.  The same is seen in other XRD graphs.

·         There is no label “a” and “b” in the figures

·         The authors mentioned a formation of surface layer in Figure 2. Why was this layer formed, which is the composition?

·         Figure 3 was obtained by TEM , it should be mention in the figure caption.

Round 2

Reviewer 3 Report

Current form of the manuscript now is good enough for publication in journal.

Reviewer 4 Report

The authors addressed my concern. The paper can be accepted for publication in Materials